# Occurrence of *Aspergillus* and *Penicillium* Species, Accumulation of Fungal Secondary Metabolites, and qPCR Detection of Potential Aflatoxigenic *Aspergillus* Species in Chickpea (*Cicer arietinum* L.) Seeds from Different Farming Systems

**DOI:** 10.3390/foods14152610

**Published:** 2025-07-25

**Authors:** Mara Quaglia, Francesco Tini, Emina Bajrami, Erica Quadrini, Mariateresa Fedeli, Michael Sulyok, Giovanni Beccari, Lorenzo Covarelli

**Affiliations:** 1Department of Agricultural, Food and Environmental Sciences, University of Perugia, 06121 Perugia, Italy; mara.quaglia@unipg.it (M.Q.); francesco.tini@unipg.it (F.T.); bajramiemina@yahoo.it (E.B.); erica-quadrini@hotmail.it (E.Q.); mtfedeli@hotmail.com (M.F.); lorenzo.covarelli@unipg.it (L.C.); 2Department of Agricultural Sciences, Institute of Bioanalytics and Agro-Metabolomics, BOKU University, 3430 Tulln, Austria; michael.sulyok@boku.ac.at

**Keywords:** aflatoxins, mycobiota, mycotoxins, mycotoxigenic fungi, organic farming system, post-harvest, pulses

## Abstract

The European chickpea market raises concerns about health risks for consumers due to contamination by mycotoxins. Contamination levels can vary depending on the farming system, and rapid and reliable screening tools are desirable. In this study, marketed chickpea seed samples from organic and non-organic farming systems were analyzed for fungal and mycotoxin contamination. *Aspergillus* and *Penicillium* were the most frequently identified mycotoxigenic genera. Significant differences in fungal detection were observed among the three isolation methods used, whose combined application is proposed to enhance detection efficiency. The number of *Aspergillus* and *Penicillium* colonies was significantly higher in the organic samples. Molecular analysis identified different species within each genus, including several not previously reported in chickpea, as well as potentially aflatoxigenic species such as *A. flavus*/*oryzae* and *A. parasiticus*. LC-MS/MS analysis revealed aflatoxin production only by *A. parasiticus*, which was present in low amounts. However, the presence of potentially aflatoxigenic *Aspergillus* species suggests that chickpeas should be monitored to detect their safety and subsequently protect consumer health. A qPCR protocol targeting the *omt-1* gene, involved in aflatoxin biosynthesis, proved to be a promising rapid tool for detecting potentially aflatoxigenic *Aspergillus* species.

## 1. Introduction

Chickpea (*Cicer arietinum* L.) is a legume of the *Fabaceae* family grown on all five continents, with a global production of about 16.5 million tons of dry grain in 2023 [1]. World production is dominated by Asia, particularly India, which is the world’s leading chickpea producer, with approximately 12.3 million tons in 2023. In Europe, chickpea production has increased in recent years, rising from 67,000 tons in 2014 to approximately 540,000 tons in 2023. In Italy, chickpea production, which was around 40,000 tons in the 1960s, drastically declined over time, reaching a minimum of approximately 4000 tons in the 1990s. However, production has begun to rise again, exceeding 33,000 tons in 2017 [1].

Like other pulse crops, chickpea is considered an important food due to its nutritional value, primarily linked to its high protein content (17–22% of dry mass) [2,3]. Chickpea is also a good dietary source of carbohydrates (60–70% dry weight), fiber, minerals, and vitamins [3]. Its consumption also provides a range of bioactive compounds with health benefits, including antioxidant, anti-diabetic, anti-cancer, and anti-inflammatory effects, which help slow the progression and reduce the incidence of chronic diseases [3,4,5].

Within *C. arietinum* species, two distinct types, Desi and Kabuli, can be distinguished based on the morphological features of the seed, flower, and protein content [3,6]. The Desi type is mainly cultivated in Africa and Asia, covering more than 80% of global chickpea production. The remaining production corresponds to the Kabuli type, primarily cultivated in Europe [3,7,8]. In Italy, the Kabuli type is the only one grown and distributed on a large scale, while the Desi type is imported and available in small, specialized markets.

Chickpea is susceptible to numerous fungal pathogens that can compromise both the quantity and quality of its production. These pathogens can infect chickpea plants from the field to the post-harvest stage and can attack several parts of the plant, including seeds [9]. Some of the most commonly associated fungi with chickpea seeds belong to the mycotoxigenic genera *Alternaria*, *Aspergillus*, *Cladosporium*, *Fusarium*, and *Penicillium* [10,11,12,13,14]. Mycotoxigenic fungi can start to colonize chickpea seeds in the field, and improper harvesting, transportation, storage, and processing techniques can further increase their presence [10]. Contamination by these fungi has also been detected in processed chickpea products, such as chickpea flour [15]. As a result, mycotoxin contamination of chickpea seeds and their derived products may pose a health risk to the final consumer. Aflatoxins and ochratoxin A (OTA) are among the most frequently reported mycotoxins in chickpea seeds and derived products, although *Fusarium* and *Alternaria* mycotoxins have also been detected [10,13,14,16]. Contamination by mycotoxigenic fungi and mycotoxins is poorly investigated in chickpeas grown in Europe, in which fungi of the mycotoxigenic *Aspergillus* and *Penicillium* genera, their related mycotoxins, such as aflatoxins B1, B2, G1, and G2, and OTA have been detected [10,17]. Currently, European legislation does not establish limits on the presence of mycotoxins in chickpeas intended as raw material or derived products [18,19]. However, the aforementioned evidence highlights the need to closely examine mycotoxin levels in this pulse product.

In Europe, chickpea can be produced either through organic or integrated farming systems, which differ, among others, also in the methods adopted for disease management, with the organic system prohibiting the use of synthetic pesticides and the integrated system allowing for their regulated use, along with the application of other protection means [20,21]. On the other hand, in some extra-European chickpea-producing areas, a conventional farming system, with pesticides as the main control means, is largely adopted. As reported for other crops, such as cereals, farming systems can affect the level of mycotoxin contamination [22].

In this context, and with the aim of verifying the impact of the farming system on the infection by mycotoxigenic fungi and on mycotoxin contamination, commercial chickpea seed samples from both organic and non-organic (integrated or conventional) farming systems were used to (a) analyze the mycobiota associated with this matrix using three different isolation methods; (b) molecularly identify the isolated *Aspergillus* and *Penicillium* species; (c) monitor the possible presence of fungal secondary metabolites in seeds; (d) assess secondary metabolite production in vitro by representative *Aspergillus* isolates; and (e) apply a qPCR assay to quantify potentially aflatoxigenic *Aspergillus* species in the seeds.

## 2. Materials and Methods

### 2.1. Marketed Chickpea Seed Samples

Marketed dried Kabuli-type chickpea seed samples were analyzed in this study, as this is the only variety cultivated and widely distributed on a large scale in Italy. A total of 20 samples were purchased in 2019 from farms or grocery stores in the Perugia area (Umbria, Central Italy) and initially selected only based on the farming system (organic, non-organic) (Table 1). Within both organic and non-organic samples, further distinctions emerged from the product label, relating to their geographical origin (Table 1). Specifically, 18 samples were from Europe (16 from Italy, 1 from Spain, and 1 from an unknown EU location), 1 sample had an extra-EU origin, while another had no origin indication. Greater homogeneity emerged regarding packaging. Indeed, all 20 samples weighed 500 g each and were packed in sealed plastic bags, except for sample 5, which was packed in an unsealed cardboard box (Table 1).

### 2.2. Isolation and Molecular Identification of the Fungal Microorganisms Belonging to the Aspergillus and Penicillium Genera Associated with the Marketed Chickpea Seed Samples

For each of the 20 chickpea seed samples, fungal isolation was carried out following the applied analysis strategy previously described [23]. A total of 150 randomly selected seeds (approximately 55 g) were used per sample. Fifty seeds were employed for the deep-freezing blotter (DFB) test [24]. For this test, the chickpea seeds were placed in Petri dishes (diameter 9 cm) on two sheets of sterile filter paper (Whatman N.1, Maidstone, UK) wetted with 10 mL of sterile deionized water. The plates were incubated for 24 h at 21 ± 2 °C in the dark for the following 24 h at −20 °C, then for 7 days at 21 ± 2 °C in the dark. Soaking and freezing allowed for the inhibition of seed germination, preventing interference with the subsequent fungal isolation. The remaining 100 seeds were placed in 9 cm diameter Petri dishes (Nuova Aptaca, Canelli, Italy) containing Potato Dextrose Agar (PDA, Biolife Italiana, Milan, Italy), pH 5.7. Of these, 50 seeds were placed on PDA without surface disinfection (PDA ND) and the remaining 50 seeds were placed on PDA following surface disinfection (PDA D). Surface disinfection was performed by immersing the seeds in a sterile aqueous solution containing 10% (*v*/*v*) ethanol (96%, Sigma-Aldrich, Saint Louis, MO, USA) and 8% (*v*/*v*) sodium hypochlorite (7–9%, Carlo Erba Reagent, Milan, Italy) for 40 s, followed by a 40 s rinse in sterile deionized water [25]. For each sample and method (DFB, PDA ND, and PDA D), five Petri dishes (replicates), each containing 10 seeds, were used, for a total of 50 seeds per method per sample. After seven days of incubation at 21 ± 2 °C in the dark, the fungal colonies developed from the seeds were observed to assess colony features. In addition, stereomicroscope (SZX9, Olympus, Tokyo, Japan) and light microscope (Axiophot, Zeiss, Jena, Germany) observations were performed to examine fungal conidophores and/or conidia (if present) to morphologically identify the fungal genera to which they belonged. Following a previously described method [23,26], each colony selected as representative of a well-defined fungal “morphotype”, based on color, morphology, growth, and microscopic features, was transferred onto new PDA plates and incubated at 21 ± 2 °C in the dark for 7 days. From the fungal colonies obtained from the 20 chickpea seed samples, this selection process resulted in 33 representative isolates, each corresponding to a distinct morphotype within the genera *Aspergillus* and *Penicillium*. Attention was focused on these two fungal genera because they were the most frequent among the mycotoxigenic fungal genera collected from the analyzed samples.

Monosporic cultures of each of these 33 representative isolates (2, 6, 33bis, 43, 44, 62, 71, 85, 89, 93, 95, 100, 112, 113, 114, 122, 126, 130, 135, 137, 141, 142, 145, 146, 148, 150, b10, b40, b50, b60, b63, b71, b77) were obtained and cultured at 21 ± 2 °C in the dark on an orbital shaker at 130 rpm in Czapek Yeast Broth (CYB) medium. The CYB medium was prepared by Czapek Dox Broth (Panrear Quimica SA, Barcelona, Spain) and Yeast Extract (Difco, Bacton Dickinson and Company, Le Pont-de-Claix, France) [26,27,28]. After 14 days of incubation, the mycelium of each fungal colony was collected by filtration through sterile filter paper (Whatman N.1, Maidstone, UK), transferred to 0.5 mL plastic tubes (Eppendorf, Hamburg, Germany), and used for DNA extraction. DNA was extracted from 0.01 g of freeze-dried (by Heto Power Dry LL3000, Thermo Fisher Scientific, Waltham, MA, USA) mycelia as previously described [24]. The monosporic cultures of each representative isolate were also stored in the mycological collection of the Plant Pathology Area, Research Unit of Plant Protection, Department of Agricultural, Food and Environmental Sciences of the University of Perugia (Perugia, Italy) and stored at −80 °C. To molecularly identify the 33 representative isolates, a phylogenetic analysis was performed using the combined sequences of the *β-tubulin* (*BenA*) and *calmodulin* (*CaM*) genes [23,29,30]. The sequences were obtained with the following primer pairs:for *BenA* gene: Bt2a (Fw) 5′-GGTAACCAAATCGGTGCTGCTTC-3′ and Bt2b (Bw) 5′- ACCCTCAGTGTAGTGACCCTTGGC-3′ [31];for *CaM* gene: CMD5 (Fw) 5′-CCGAGTACAAGGARGCCTTC-3′ and CMD6 (Bw) 5′-CCGATRGAGGTCATRACGTGG-3′ [32].

PCR reactions were carried out as previously described [23,26] with slight modifications. Specifically, these were performed in a volume of 50 µL, including 2 µL of a DNA working solution (25 ng/µL), for a total of 50 ng of template per reaction.

The reactions were performed using a T100TM Thermal Cycler (Biorad, Foster City, CA, USA) and the following cycling profiles [33]:For *BenA*: initial denaturation at 95 °C for 4 min, followed by 35 cycles of 94 °C for 1 min, 65 °C for 1 min and 72 °C for 1 min, with a final extension at 72 °C for 8 min;For *CaM*: initial denaturation at 95 °C for 4 min, followed by 35 cycles of 95 °C for 1 min, 55.5 °C for 1 min and 72 °C for 2 min, with a final extension at 72 °C for 8 min.

Amplification products were separated on a 2% agarose gel in 1 × TAE buffer with RedSafe^TM^ (4% *v*/*v*) (Chembio, Medford, NY, USA) at 110 V for 30 min and subsequently sent to Genewiz Europe (Azenta Life Sciences, Leipzig, Germany) for purification and sequencing. Each consensus sequence was examined using the MEGA software version 7.0 [34] through the View Sequencer File (Trace Editor) functionality. The sequences were then compared with those deposited in the National Centre for Biotechnology Information (NCBI) Basic Local Alignment Search Tool (BLAST) database [35] and finally deposited in GenBank (Appendix A). Phylogenetic analyses were performed using the above-reported MEGA software version 7.0 and a combined dataset of *BenA* and *CaM* gene sequences of the *Aspergillus* (Appendix A) and *Penicillium* (Appendix A) isolates obtained in this study, along with representative isolates found in GenBank, including those of the main species reported on *C. arietinum* (Appendix A). According to a previously described method [30], for a more precise identification of isolates that clustered in the *Nigri* section, additional representative *Aspergillus* species belonging to this section were obtained from GenBank and included in the analyses (Appendix A). As outgroups, *Talaromyces flavus* CBS 310.38 and *Talaromyces marneffei* CBS 388.87 were included in the phylogenetic analyses of *Aspergillus* [36,37] (Appendix A) and *Penicillium* [38] (Appendix A). Additionally, *Talaromyces purpurogenus* CBS 286.36 was included as an outgroup for the phylogenetic analyses of *Penicillium* [37] (Appendix A). After concatenation, *BenA* plus *CaM* gene sequences were aligned, and nucleotide gaps and missing data were removed. Phylogenetic trees were built using the Neighbor-Joining method [39], with a bootstrap test based on 1000 replicates [40]. Evolutionary distances were computed using the Maximum Composite Likelihood method [41].

### 2.3. Analysis of Fungal Secondary Metabolites in the Marketed Chickpea Seed Samples and Determination of Secondary Metabolite Profile Produced In Vitro by Selected Fungal Isolates

For each sample, 50 g of seeds was finely ground using a GM200 mill (Retsch, Verder Scientific srl, Pedrengo, Italy), and 5 g of the seed powder was subjected to analysis of fungal secondary metabolites. The detection of fungal secondary metabolites was performed using liquid chromatography–tandem mass spectrometry (LC-MS/MS) following a previously described method [42], which enables the identification and quantification of a wide range of fungal secondary metabolites (over 700, including main and emerging mycotoxins). Confirmation of positive metabolite identification was obtained by acquiring two MS/MS signals per analyte, which yields 4.0 identification points, according to the European Commission [43]. In addition, retention time and ion ratio had to agree with the related values of authentic standards within 0.03 min and 30%, respectively. The accuracy of the method was verified continuously by participation in a proficiency-testing scheme (BIPEA, Gennevilliers, France), with >96% of the 2400 results submitted so far (including chickpea samples) exhibiting a z-score of −2 < z < 2. The analysis was carried out on a 5 g subsample from each of the 20 marketed chickpea seed samples.

To assess the ability to produce aflatoxins of six potentially aflatoxigenic isolates, five molecularly identified isolates belonging to the *A. flavus*/*oryzae* species and one molecularly identified as *A. parasiticus*, their secondary metabolite profile was determined on 5 g of freeze-dried (by Heto Power Dry LL3000, Thermo Fisher Scientific, Waltham, MA, USA) cultures following a previously described method [42]. These seven isolates were previously grown for 10 days at 21 ± 2 °C in the dark on Czapek Yeast Autolysate (CYA) agar medium (HiMedia Laboratories, GmbH, Einhausen, Germany), as already described [23,44].

### 2.4. DNA Quantification of Aflatoxigenic Aspergillus Species in Chickpea Seed Samples by qPCR

To evaluate the DNA amounts of aflatoxigenic *Aspergillus* species in chickpea seed samples, a qPCR method was used.

For the realization of the standard curve for the quantification of fungal DNA accumulation, the aflatoxigenic strain A6 of *A. flavus*/*oryzae*, previously isolated from date fruits and phylogenetically characterized, was used [26]. The A6 strain was grown on sterile cellophane disks placed on PDA in Petri dishes (9 cm diameter) at 21 ± 2 °C in the dark. After seven days, the mycelium was scraped, freeze-dried by Heto Power Dry LL3000 (Thermo Fischer Scientific, Waltham, MA, USA) and finely ground by a MM400 mill (Retsch, Verder Scientific srl, Pedrengo, Italy) at 25 Hz for 6 min.

For each chickpea sample, a sub-sample of 50 mg taken from the 50 g of seed powder previously prepared for LC-MS/MS analysis was used for DNA extraction.

Genomic DNA isolation from isolate A6, as well as from 50 mg of each chickpea sample, was performed using the ZR Fungal/Bacterial DNA Kit^TM^ (Zymo Research, Irvine, CA, USA), according to the manufacturer’s protocol. Quality and concentration of the DNA were assessed by QubitTM 3.0 fluorometer (Thermo Fisher Scientific, Waltham, MA, USA).

A serial dilution ranging from 20 ng to 0.02 pg of *A. flavus* DNA and from 70 ng to 0.07 pg of chickpea seed DNA, with a 10-fold dilution factor, was used to plot standard curves, performing two technical replicates per concentration, by using a CFX96 Opus 96 Real-Time PCR System (Bio-Rad, Hercules, CA, USA). The primer pairs used for DNA quantification by qPCR were
For aflatoxigenic *Aspergillus* species, F-omt (5′-GGCCGCCGCTTTGATCTAGG-3′) and R-omt (5′-ACCACGACCGCCGCC-3′), designed on the *omt-1* gene coding for the sterigamatocystin *O*-methyltransferase, a key enzyme in the aflatoxin biosynthetic pathway, which generates an amplicon of 123 bp and has previously been used for the detection of *A. flavus* and *A. parasiticus* [45,46];For chickpea DNA assay, Fw (5′-CCAAGGTCAAGATCGGAATCA-3′) and Rev (5′-CAAAGCCACTCTAGCAACCAAA-3′), designed on the internal control gene *Glyceraldehyde-3-phosphate dehydrogenase* (*GADPH*), generating an amplicon of 65 bp [47].

The qPCR cycle was composed of an initial step at 50 °C for 2 min and 95 °C for 10 min, followed by 40 cycles at 95 °C for 15 s, 61 °C for 1 min, heating at 95 °C for 10 s, cooling at 65 °C, and a final temperature increase to 95 °C at a rate of 0.5 °C every 5 s, with the measurement of fluorescence. A dissociation curve was included at the end of the qPCR program to detect potential primer-dimers and nonspecific amplification products. Each reaction was performed in a total volume of 12.5 µL, containing 6 µL of SYBR^TM^ Select Master Mix CFX (Thermo Fisher Scientific, Waltham, MA, USA), 1.5 µL of each 2 µM primer, 0.5 µL of sterile deionized water, and 2.5 µL of template DNA. Standard curves were generated by plotting the logarithmic values of the above-reported fungal or chickpea DNA amounts versus the corresponding cycles of threshold (Ct) values. The line equations, R^2^ values, reaction efficiencies, and limits of detection (LODs) were calculated. The accumulation of *Aspergillus* DNA in chickpea was expressed as the ratio of the detected DNA (pg) to the total chickpea seeds DNA (ng).

### 2.5. Statistical Analysis

The data on the isolation frequency of the different fungal genera obtained from the 20 chickpea seed samples, included as a whole, were subject to one-way analysis of variance (ANOVA) and the “isolation method” (PDA D, PDA ND or DFB) or the “farming system” (organic or non-organic) were included as variables. Additionally, the data on the frequency of different fungal genera obtained from each single chickpea seed sample were subject to one-way ANOVA, including the “sample” as a variable. All statistical analyses were realized using the extension for Excel^®^ “DSAASTAT” version 1.514 [48]. In the case of a significant ANOVA (*p* < 0.05), Tukey’s Honestly Significant Difference (HSD) multiple-comparison test was performed to determine significant differences between the means. Correlation between the amount of isolation of potentially aflatoxigenic *Aspergillus* species and the quantification of their DNA level in the marketed chickpea seed samples by qPCR was assessed by using the Pearson (*r*) correlation coefficient.

## 3. Results

### 3.1. Fungal Microorganisms Associated with the Marketed Chickpea Seed Samples

After one week of incubation, fungal colonies developed from all chickpea samples (100%), with a total of 1615 colonies obtained (considering all the isolation methods used as a whole). Based on morphological features, 1005 colonies (62.2% of the total) were identified as belonging to the *Aspergillus* genus, 231 colonies (14.3%) to the *Rhizopus* genus, 221 colonies (13.7%) to the *Penicillium* genus, 51 colonies (3.2%) to the *Cladosporium* genus, and, finally, 16 colonies (1.0%) to the *Alternaria* genus. For the remaining 91 colonies (5.6% of the total), it was impossible to identify a genus based on morphological characteristics only, so these colonies were classified as “other”. Figure 1 summarizes the total number of colonies of the different fungal genera obtained with all the isolation methods (PDA D, PDA ND, and DFB) from the marketed chickpea seed samples.

*Aspergillus*, *Penicillium*, *Cladosporium*, and *Alternaria* are well-known mycotoxigenic genera. Among them, *Aspergillus* was the most widespread, being detected in 19 out of the 20 samples (*Aspergillus* was not detected only in sample 14) (Table 2). *Penicillium* was the second most frequently detected mycotoxigenic genus, present in 13 samples (Table 2). *Cladosporium* was detected in four samples, while *Alternaria* was found in three samples only (Table 2). The co-occurrence of *Aspergillus*, *Penicillium*, *Cladosporium*, and *Alternaria* colonies was observed in two samples (Table 2). However, the co-presence of three different mycotoxigenic genera was also detected. For instance, in two samples, *Aspergillus*, *Penicillium*, and *Cladosporium* co-occurred, while in another sample, *Aspergillus*, *Penicillium,* and *Alternaria* were detected simultaneously (Table 2).

The three different isolation methods used in this study (PDA D, PDA ND, or DFB) allowed us to obtain a significantly different number of total fungal colonies, as well as of some fungal genera (Figure 2). In detail, the PDA ND method allowed us to obtain a significantly (*p* < 0.05) higher number of total fungal colonies (6.69 ± 0.32) compared to those obtained using the PDA D (4.85 ± 0.41) and DFB (4.85 ± 0.32) methods (Figure 2). Similarly, the PDA ND method allowed us to obtain a significantly (*p* < 0.05) higher number of *Aspergillus* (4.45 ± 0.3) and *Rhizopus* (1.56 ± 0.25) colonies than the PDA D (3.29 ± 0.35 and 0.49 ± 0.17, respectively) and DFB (2.31 ± 0.25 and 0.26 ± 0.08, respectively) methods (Figure 2). Conversely, the PDA ND method allowed us to obtain a significantly (*p* < 0.05) lower number of *Cladosporium* colonies (0) compared to those obtained using the DFB method (0.42 ± 0.14) (Figure 2). No significant differences (*p* > 0.05) were observed among the three isolation methods in the number of detected *Penicillium* or *Alternaria* colonies (Figure 2).

The farming system (organic or non-organic) adopted to produce the marketed chickpea seed samples also allowed us to obtain a significantly different number of total fungal colonies, as well as of some fungal genera (Figure 3). In detail, the marketed chickpea seed samples obtained with the organic farming system showed a significantly (*p* < 0.05) higher number of total fungal colonies, as well as of *Aspergillus*, *Penicillium*, and *Cladosporium* colonies, than the samples obtained with the non-organic farming system. Conversely, the marketed chickpea seed samples obtained with the non-organic farming system showed a significantly (*p* < 0.05) higher number of *Rhizopus* colonies than the samples obtained with the organic farming system. Finally, no significant differences were detected between the two farming systems in the number of isolated *Alternaria* colonies (Figure 3). Appendix A summarizes the average number of total fungal colonies or fungal colonies belonging to the *Aspergillus*, *Penicillium*, *Cladosporium*, *Alternaria*, and *Rhizopus* genera or other genera developed from each single marketed chickpea seed sample.

### 3.2. Molecular Identification of the Fungal Isolates Belonging to the Aspergillus and Penicillium Genera Associated with the Marketed Chickpea Seed Samples

As explained in Section 2.2, 33 isolates within the genera *Aspergillus* and *Penicillium* were selected as representative of the observed morphotypes. Fifteen isolates (44, 113, 114, 122, 126, 141, 146, 150, b10, b40, b50, b60, b63, b71, b77) were morphologically identified as *Penicillium*, whereas the remaining eighteen isolates (2, 6, 33bis, 43, 62, 85, 89, 93, 95, 100, 112, 130, 135, 137, 142, 145, 146, 148) were morphologically identified as *Aspergillus*. BLAST analysis of *BenA* and *CaM* gene sequences confirmed the identification of the isolates, also defining the species.

As reported previously [38,49], using the concatenated sequences of *BenA* and *CaM* genes, 10 major clades emerged in the phylogram of both *Aspergillus* (Figure 4) and *Penicillium* (Figure 5) species.

In the phylogram of the *Aspergillus* species (Figure 4), clade A included species of the section *Nigri*, clade B of the section *Fumigati*, clade C of the section *Flavi*, clade D of the section *Usti*, clade E of the section *Nidulantes*, clade F of the section *Versicolores*, clade G of the sections *Candidi*/*Circumdati*, clade H of the section *Flavipedes*, clade I of the section *Terrei,* and clade J of the section *Cremei*. In addition, within clade A, two subclades, here named *tubingensis* and *awamori*/*welwitschiae*, were identified. Isolates 100, 112, 130, 135, 142, and 145 clustered in the subclade *tubingensis,* together with isolates NRRL 4875 and CBS 134.48 of *A. tubingensis*, while isolates 6, 85, and 95 clustered in the subclade *awamori*/*welwitschiae* together with isolates CBS 557.65 of *A. awamori* and CBS 139.54 *of A. welwitschiae*. Isolates 2, 43, 62, 71, 89, 137, and 148 clustered in clade C, with isolate 137 grouping with isolate CBS 103.14 of *A. tamarii*, isolate 148 with isolate CBS 100926 of *A. parasiticus,* and isolates 2, 43, 62, 71, and 89 with isolates CBS 569.65 and CBS 100927 of *A. flavus* and CBS 102.07 of *A. oryzae*. Finally, isolate 93 clustered in clade D together with isolate CBS 756.74 of *A. pseudodeflectus*, while isolate 33 bis clustered in clade F with isolate CBS 593.65 of *A. sidowii*.

In the phylogram of the *Penicillium* species (Figure 5), clade A included species of the section *Fasciculata*, clade B of the section *Penicillium*, clade C of the section *Digitata*, clade D of the section *Chrysogena*, clade E of the section *Canescentia*, clade F of the section *Brevicompacta*, clade G of the section *Aspergilloides*, clade H of the section *Exilicaulis*, clade I of the section *Lanata*-*Divaricata,* and clade J of the section *Citrina*. Isolates 113, 141, b60, b63, and b77 clustered in clade A, with isolate 113 grouped with isolate IMI 92917 of *P. crustosum*, isolate 141 with isolate CBS 222.28 of *P. polonicum*, isolate b60 with isolates F 759 and F 727 of *P. cellarum*, isolate b63 with isolate CV 1331 of *P. melanoconidium,* and isolate b77 with isolate CBS 390.48 of *P. viridicatum*. Isolate 146 clustered in clade B, together with isolate CBS 325.48 of *P. expansum*; isolates 122, b10, b40, and b60 clustered in clade D together with isolate CBS 306.48 of *P. chrysogenum*; isolate 44 clustered in clade E together with isolate CBS 300.48 of *P. canescens*. Within clade F, isolate 150 clustered with isolate CBS 232.60 of *P. olsonii*, while isolate 114 clustered together with isolates NRRL 2011, NRRL 2012, and CBS 257.29 of *P. brevicompactum*. Finally, isolate b71 clustered in clade G together with isolates CBS 125543 and NRRL 35684 of *P. glabrum,* and isolate 126 clustered in clade H together with isolates CBS 31248 and CBS 33070 of *P. corylophilum*.

Within the *Aspergillus* genus (Figure 6a), *A. flavus*/*oryzae* (533 colonies), and *A. tubingensis* (289 colonies) were the most abundant species, followed by *A. awamori*/*welwitschiae* (79 colonies), *A. sidowii* (45 colonies), *A. pseudodeflectus* (40 colonies), *A. parasiticus* (17 colonies), and *A. tamarii* (2 colonies). Most of the detected *Aspergillus* species were obtained using all the isolation methods (PDA D, PDA ND, and DFB), except for *A. parasiticus* and *A. tamarii,* which were obtained only using the PDA ND method (Figure 6a).

Within the *Penicillium* genus (Figure 6b), *P. brevicompactum* (93 colonies) was the most abundant species, followed by *P. cellarum* (43 colonies), *P. corylophilum* (21 colonies), *P. chrysogenum* (19 colonies), *P. crustosum* (16 colonies), *P. glabrum* (11 colonies), *P. melanoconidium* (8 colonies), *P. viridicatum* (6 colonies), *P. expansum* (2 colonies), and *P. olsonii* and *P. canescens* (1 colony each). Only three of the eleven *Penicillium* species (*P. brevicompactum*, *P. corylophilum*, and *P. crustosum*) were isolated using all three methods. *P. crustosum* and *P. cellarum* were obtained with two of the three isolation methods, with *P. crustosum* detected with PDA D and PDA ND and *P. cellarum* with PDA D and DFB. *P. glabrum*, *P. melanoconidium,* and *P. viridicatum* were detected only by DFB, while *P. expansum*, *P. canescens*, and *P. olsonii* were detected only using PDA D.

Finally, focusing on the potential aflatoxigenic *Aspergillus* species (*A. flavus*/*oryzae* and *A. parasiticus*), significant differences (*p* < 0.05) were observed comparing the presence of these species between the two farming systems. In detail, a significantly (*p* < 0.05) higher number of colonies belonging to potential aflatoxigenic species was isolated from marketed chickpea seed samples obtained with the organic farming system than from those obtained with the non-organic farming system (Figure 7).

### 3.3. Fungal Secondary Metabolites in Marketed Chickpea Seed Samples and in In Vitro Cultures of Potentially Aflatoxigenic Aspergillus Isolates

All the secondary metabolites detected by LC-MS/MS analysis in at least one marketed chickpea seed sample are listed in Table 3. Among these, the *Aspergillus* metabolite sydowinin B was detected (at levels ranging from 0.76 to 3.94 ng/g) in all samples. In sample 6, sporogen AO, another *Aspergillus* metabolite, was identified at a concentration of 0.19 ng/g, as well as 3-nitropropionic acid in sample 13 at 0.88 ng/g. Tenuazonic acid (6.35 and 6.50 ng/g in samples 1 and 13, respectively) and tentoxin (0.30 ng/g in sample 3), both produced by *Alternaria* spp., were also detected. In 19 out of the 20 samples, the *Fusarium* metabolite enniatin B was detected in concentrations ranging from 0.01 to 0.72 ng/g. In some samples, enniatin B showed co-occurrence with other *Fusarium* metabolites, such as enniatin A (0.01 ng/g), enniatin A1 (0.01–0.04 ng/g), enniatin B1 (0.01–0.21 ng/g), destruxin B (0.01–0.42 ng/g), and W493 (0.11–0.29 ng/g). Other typical *Fusarium* secondary metabolites, such as moniliformin and deoxynivalenol, were detected in samples 1, 2, 3, and 4 (0.14–0.56 ng/g) and 1, 3, 4, and 5 (0.46–2.67 ng/g), respectively.

All the secondary metabolites produced in vitro by six potentially aflatoxigenic *Aspergillus* isolates and detected by LC-MS/MS are listed in Table 4. Five out of the six isolates analyzed (2, 43, 62, 71, and 89) were identified as *A. flavus*/*oryzae*, and one (148) as *A. parasiticus*, based on the previous phylogenetic analysis. Only isolate 148 (*A. parasiticus*) produced aflatoxins (aflatoxicol, aflatoxin B1, B2, G1, G2, and M1) and their precursors O-methylsterigmatocystin and sterigmatocystin. Isolate 148 also produced averufin, norsolorinic acid, aspergillicin derivatives, helvolic acid, and the compounds versiconal acetate and versicolorin C (Table 4). Kojic acid and 3-nitropropionic acid were produced by all seven isolates, with higher amounts detected in isolate 148 (*A. parasiticus*) (Table 4). The fungal metabolites asperfuran, heptelidic acid, gliocladic acid, penicillin G, sporogen AO, cyclopiazonic acid, and NP 1243 were exclusively produced by isolates of *A. flavus*/*oryzae* (Table 4). Aspergillicin derivatives were biosynthesized by isolate 2 (*A. flavus*/*oryzae*) and 148 (*A. parasiticus*) (Table 4). Finally, the isoacumarin dichlorodiaportin was produced by isolates of *A. flavus*/*oryzae*, and *A. parasiticus*, while its methylated form, O-methyldichlorodiaportin, was detected only in isolate 137 148 (*A. parasiticus*) (Table 4).

### 3.4. Potentially Aflatoxigenic Aspergillus Species Quantified by qPCR in the Marketed Chickpea Seed Samples and Comparison of qPCR with Isolation Results

The R^2^ value and the efficiency of qPCR reactions were 0.99 and 97% for chickpea seed samples and 0.99 and 98% for *A. flavus*, respectively. For the quantification of potentially aflatoxigenic *Aspergillus* DNA, the LOD was 0.5 pg.

By qPCR, variable amounts of potentially aflatoxigenic *Aspergillus* DNA were detected in 15 out of 20 marketed chickpea seed samples. Among these, nine samples (1–5 and 7–10) were produced with the organic farming system, whereas the remaining six (11, 13, 15, 16, 18 and 19) with the non-organic farming system. The higher amount of potentially aflatoxigenic *Aspergillus* DNA was detected in sample 1 (Figure 8a). Comparing these qPCR results with those of the isolation and molecular identification of the fungal isolates, potentially aflatoxigenic *Aspergillus* species were obtained from samples 1–7, 10–13, 15, 16, 18–20. Notably, sample 3 showed the highest presence of *A. flavus*/*oryzae* with the isolation methods, while sample 19 was the only one positive for *A. parasiticus* with the isolation methods (Table 5). In contrast, no aflatoxigenic species were detected neither by isolation nor by qPCR in samples 14 and 17 (Figure 8a and Table 5). Interestingly, qPCR assays detected variable amounts of *Aspergillus* DNA in samples 8, 9, and 10, but no potentially aflatoxigenic *Aspergillus* species were isolated from these samples. Conversely, *Aspergillus* DNA was not detected in samples 6, 12, and 20, despite the isolation of *A. flavus*/*oryzae* colonies from them (Figure 8a and Table 5).

The correlation analysis between the amount of isolation of the aflatoxigenic *Aspergillus* strains and the quantification of their DNA level in chickpea samples was positive with a Pearson’s coefficient (*r*) value of 0.50.

Compared to the chickpea samples from the non-organic farming system, the chickpea samples from organic farming not only gave positive results on qPCR for the presence of DNA of potentially aflatoxigenic *Aspergillus* species in a greater number of cases but also presented a higher amount of DNA of aflatoxigenic species, although no statistically significant difference (*p* > 0.05) was found (Figure 8b).

## 4. Discussion

The composition of macro- and micronutrients, along with bioactive compounds, makes chickpea a healthy food with beneficial effects on humans [2,5]. However, from farm to fork, contamination by mycotoxigenic fungi and their secondary metabolites can negatively affect chickpea safety for final consumers. Indeed, fungal contamination by mycotoxigenic genera such as *Alternaria*, *Aspergillus*, *Cladosporium*, *Fusarium*, and *Penicillium* has been reported in chickpea and chickpea-derived products in several chickpea-growing areas [9,10,11,12,15,57,58,59,60,61,62,63,64,65,66,67,68].

As shown in previous reports, contamination by fungal species belonging to well-known mycotoxigenic genera such as *Aspergillus*, *Penicillium*, *Cladosporium*, and *Alternaria* was also detected in the present survey. In detail, considering the three isolation methods used, *Aspergillus* was the most frequently identified genus, detected in 19 out of 20 analyzed samples. In some cases, multiple fungal isolates were obtained from chickpea seeds of a single sample and occasionally also from the same seed. These isolates belonged to one or more of the above-mentioned fungal genera, with *Penicillium* being the most common genus, along with *Aspergillus*. The simultaneous infection of food matrices by different mycotoxigenic fungi and the possible consequent simultaneous contamination with their secondary metabolites is a long-standing concern that has been considered by both the World Health Organization (WHO) and the European Food Safety Authority (EFSA) [69].

Among the two most widespread mycotoxigenic genera (*Aspergillus* and *Penicillium*) detected in the analyzed samples, molecular and phylogenetic analyses allowed for the identification of several species. Within the genus *Aspergillus*, seven species were identified: *A. awamori*/*welwitschiae*, *A. flavus*/*oryzae*, *A. parasiticus*, *A. pseudoflectus*, *A. tamarii*, *A. tubingensis*, and *A. sidowii*. Within the *Nigri* section, the two cryptic phylogenetic species *A. awamori* and *A. welwitschiae*, initially synonymized [70], have been recently reclassified as synonyms of *A. niger* [71]. Similarly, within the *Flavi* section, comparative pangenome analysis supports the hypothesis that *A. flavus* and *A. oryzae* are the same species [72]. The species *A. niger* (including the synonym *A. ficuum*), *A. flavus*/*oryzae*, *A. parasiticus,* and *A. sydowii* have been previously reported in chickpea seed and/or chickpea flour [9,11,12,15,57,60,61,62,65,73,74,75,76]. Within the genus *Penicillium*, 11 species were identified: *P. brevicompactum*, *P. canescens*, *P. cellarum*, *P. chrysogenum*, *P. corylophilum*, *P. crustosum*, *P. expansum*, *P. glabrum*, *P. melanoconidium*, *P. olsonii*, *P. polonicum*, and *P. viridicatum*. Among these, *P. canescens*, *P. chrysogenum*, *P. expansum*, and *P. viridicatum* have been previously reported in chickpea seeds [61,76]. To our knowledge, several *Aspergillus* and *Penicillium* species identified in this study, such as *A. pseudoflectus*, *A. tamarii*, *A. tubingensis*, *P. brevicompactum*, *P. cellarum*, *P. corylophilum*, *P. crustosum*, *P. glabrum*, *P. melanoconidium*, *P. olsonii*, and *P. polonicum*, expand the range of fungal species that could be associated with chickpea seeds.

As previously observed in studies on other plant species, the isolation method significantly influences both the quantity of fungal colonies that develop from a given matrix and their quality in terms of fungal genera/species, which may be promoted by the specific method used for the mycological analysis [23,77,78]. In particular, in the present investigation, the PDA ND method allowed for the isolation of a significantly higher number of *Aspergillus* and *Rhizopus* colonies, whereas using the DFB method, a significantly higher presence of *Cladosporium* colonies was detected. Instead, no significant differences emerged among the three isolation methods regarding *Penicillium* or *Alternaria* colonies.

In addition, as previously shown [23], the differences among the isolation methods were not only quantitative but also qualitative. Indeed, within the genus *Aspergillus*, only the PDA D method allowed for the isolation of all seven identified species, whereas *A. parasiticus* and *A. tamarii* were not detected using the PDA ND or DFB methods. Similarly, within the genus *Penicillium*, no single method allowed for the isolation of all 11 identified species. Specifically, *P. glabrum*, *P. melanoconidium*, and *P. viridicatum* were detected only using the DFB method, which, however, did not allow for the isolation of *P. crustosum*, *P. expansum*, *P. olsonii,* and *P. canescens*. The latter three species, along with *P. cellarum*, were also not obtained using the PDA ND method, which allowed us to obtain only four out of the eleven identified *Penicillium* species. Therefore, as suggested by previous investigations on other food matrixes [23,77,78] and confirmed by the present study, the combined use of the three different isolation methods (PDA D, PDA ND, and DFB) is the strategy proposed here to obtain the best representation, both quantitatively and qualitatively, of the various fungal species that may be associated with a specific food matrix. This is because certain methods involving surface disinfection of the seed (for example, PDA D) may prevent the development of those fungal species that are located exclusively on the external part of the seed. At the same time, surface disinfection may favor the growth of species located inside the seed because the absence of external species allows them to develop without being outcompeted. Finally, techniques that do not include the use of artificial growth substrates rich in nutrients (for example, DFB) may allow for the development of slow-growing species, which would otherwise be overgrown by fast-growing ones in the presence of nutrient-rich media.

This study also showed a significant effect of the farming system on the number of isolated fungi, with a significantly higher presence of total fungal colonies, as well as those belonging to the genera *Aspergillus*, *Penicillium*, and *Cladosporium*, obtained from the examined organic chickpea seed samples. In particular, potentially aflatoxigenic *Aspergillus* species (such as *A. flavus*/*oryzae*, and *A. parasiticus*) were found in significantly higher numbers in organic chickpea seed samples compared to those from the non-organic farming system. A higher contamination risk by mycotoxigenic fungi and their metabolites in organic samples has recently been reported in maize [79], where a greater number of *Aspergillus*, *Penicillium*, and *Fusarium* colonies, as well as of their metabolites, mainly aflatoxins B1, B2, G1, and G2, and fumonisin B1, were detected in comparison to non-organic samples. Similarly, in a previous study [23], a higher number of *Penicillium* and *Cladosporium* colonies was reported from organic quinoa seed samples in comparison to non-organic ones. Analyzing flour samples, it has been reported that organic flour was more prone to higher fungal and mycotoxin contamination compared to non-organic flours [80]. The growing demand for organic food is driven by consumers’ belief that it is healthier and safer than food produced in non-organic farming systems, including integrated ones [81]. However, as already documented [82], this work, together with other studies from the literature reported above, shows that consumers’ perception is not always supported by experimental data.

The detection and quantification of potentially aflatoxigenic *Aspergillus* species, such as *A. flavus*/*oryzae* and *A. parasiticus*, in the 20 chickpea seed samples were also performed using a qPCR assay. The protocol was based on the use of the primer pair F-omt and R-omt, designed on the *omt-1* gene, which encodes for sterigmatocystin *O*-methyltransferase, a key enzyme in the aflatoxin biosynthetic pathway [45,46]. The specificity of this primer pair for potentially aflatoxigenic *Aspergillus* species and its inability to amplify DNA from non-aflatoxigenic *Aspergillus* species has already been demonstrated [46]. A non-perfect correspondence was observed between fungal isolation results and the detection of fungal DNA by qPCR. In fact, the correlation between the DNA level of the aflatoxigenic *Aspergillus* species and the isolation frequency of the aflatoxigenic strains was positive, albeit without a high linearity.

The low presence of aflatoxigenic *Aspergillus* species isolated in the marketed chickpea seed samples 6, 12, and 20 does not justify a negative qPCR result. In fact, in samples 8, 9, and 10, from which no colonies of potentially aflatoxigenic *Aspergillus* species were isolated, qPCR yielded positive results. In these latter samples, the discrepancy between the two detection methods can be attributed to the higher sensitivity of the molecular approach, which amplifies specific genomic regions of the target organisms, allowing for the detection of infections, even in the samples that were found to be negative on isolation. However, it is important to note that, while a sub-sample of 150 dried chickpea seeds was used for fungal isolation, only 100 mg of the sample was used for DNA extraction. Reducing the sub-sample weight also decreased the probability of detecting fungal structures (spores/mycelium), explaining why the primer pair F-omt and R-omt was not able to amplify DNA of potentially aflatoxigenic *Aspergillus* species in samples 6, 12, and 20.

The LC-MS/MS analysis carried out to evaluate the in vitro secondary metabolite profile produced by selected isolates belonging to potentially aflatoxigenic *Aspergillus* species (*A. flavus*/*oryzae*, *A. parasiticus*) showed results consistent with those reported for the three species to which the isolates under investigation belonged [30,53,83,84,85]. However, among the investigated isolates, only isolate 148 of *A. parasiticus*, obtained at low frequency from sample 19, was able to produce aflatoxins in vitro. Specifically, it produced all the main aflatoxins (B1, B2, G1, and G2), as well as aflatoxin M1 and other biosynthetically related toxic compounds, such as aflatoxicol, sterigmatocystin and its methylated form, and versicolorin, including versicolorin C. Consistently with the literature, the anthraquinoid averufin and norsolorinic acid were detected among the metabolites of isolate 148 of *A. parasiticus* [53].

*Aspergillus parasiticus* and *A. flavus* are the two main aflatoxigenic members of the *Aspergillus* section *Flavi*, with *A. parasiticus* able to produce aflatoxins of both B- (including aflatoxin M) and G- groups and *A. flavus* able to produce only those of group B [53,86]. Aflatoxins non-producing strains are uncommon in *A. parasiticus*. For example, in [87], it was reported that only 11 of 185 *A. parasiticus* isolates obtained from soil, debris, and peanut seeds in Argentina were not able to synthesize aflatoxins. Instead, within the species *A. flavus*, both aflatoxin-producing and non-producing strains are included, with the non-aflatoxigenic isolates that may be exploited for the biological control of aflatoxigenic isolates [88,89]. The ability to synthesize aflatoxins was also considered a useful characteristic to distinguish the toxigenic species *A. flavus* from the non-toxigenic species *A. oryzae* [90,91]. However, as reported above, comparative pangenome analysis supports the hypothesis that *A. flavus* and *A. oryzae* are the same species, with non-aflatoxigenic strains widely distributed across the phylogenetic tree [72].

The inability to produce aflatoxins by the potentially aflatoxigenic *Aspergillus* isolates obtained with the highest frequency (such as those belonging to the *A. flavus*/*oryzae* species), along with the low isolation frequency of the only aflatoxin-producing strain, justifies the absence of aflatoxins in the marketed chickpea seed samples surveyed in this study. Nevertheless, the identification of these species initially raised the suspicion of aflatoxin contamination. However, LC-MS/MS analysis conducted in the marketed chickpea seed samples showed the presence of other secondary fungal metabolites, some of which were related to *Aspergillus* contamination, such as 3-nitropropionic acid, sporogen AO, and sydowinin B, while others were associated with contamination by *Alternaria* and/or *Fusarium* species.

The absence of aflatoxins and of other major mycotoxins in the analyzed samples, 18 of which had a European origin (including 16 produced in Italy), could justify the lack of a maximum mycotoxin limit for chickpeas in Commission Regulation (EC) 2024/1756 [18], as well as in Commission Regulation (EC) 1881/2006 [92]. However, due to the absence of these legal limits, chickpeas are excluded from official mycotoxin contamination monitoring. As a result, in a risk-based assessment of aflatoxin B and G contamination in food and commodities imported into Italy, chickpea samples were excluded because of the lack of such regulatory limits [93]. Nonetheless, a past survey conducted by the Environmental Regional Protection Agency (ARPA) in the Piedmont region (Northern Italy) as part of a monitoring plan established by the European Food Security Authority (EFSA), which included commodities not subject to mycotoxin content regulation, detected aflatoxins in 57% of the analyzed chickpea seed samples [94]. In addition, contamination of chickpeas by aflatoxins has been reported in the literature [10]. Taken together, these data suggest that chickpeas should be considered a matrix to be monitored, especially in light of their increasing presence in the European diet, which may lead to greater human exposure to chickpea-associated contaminants, including mycotoxins. Therefore, the growing use of chickpeas should be accompanied by a higher attention to food safety through the development of rapid and reliable screening tools. The combined application of three isolation methods (PDA D, PDA ND, and deep-freezing blotter), along with a qPCR protocol based on the F-omt and R-omt primer pair, is proposed here to reveal the broad variability among *Aspergillus* and *Penicillium* species and to provide a preliminary assessment of the risk of aflatoxin contamination in chickpea seeds, respectively.

Although for many crops, a positive correlation has been demonstrated between environmental conditions, such as high temperatures and humidity, and contamination by mycotoxigenic fungi and mycotoxins during cultivation and storage, limited information is available for pulse crops, including chickpeas [16]. Moreover, growing practices, as well as harvest and post-harvest conditions and operations, can significantly affect chickpeas’ safety [95]. The protocol proposed in this study, developed using marketed dry chickpea samples and combined with LC-MS/MS analysis for fungal secondary metabolite detection, could be applied in future research on chickpea seed samples collected at different stages of the growing, drying, and storage cycles. This would allow for the assessment of the impact of various factors, such as climatic conditions, crop practices (including plant genotype, crop rotation, and phytosanitary treatments), and storage techniques (e.g., controlled atmosphere), on fungal infections and mycotoxin contaminations to identify risk factors and practices that reduce contamination levels.

## Figures and Tables

**Figure 1 foods-14-02610-f001:**
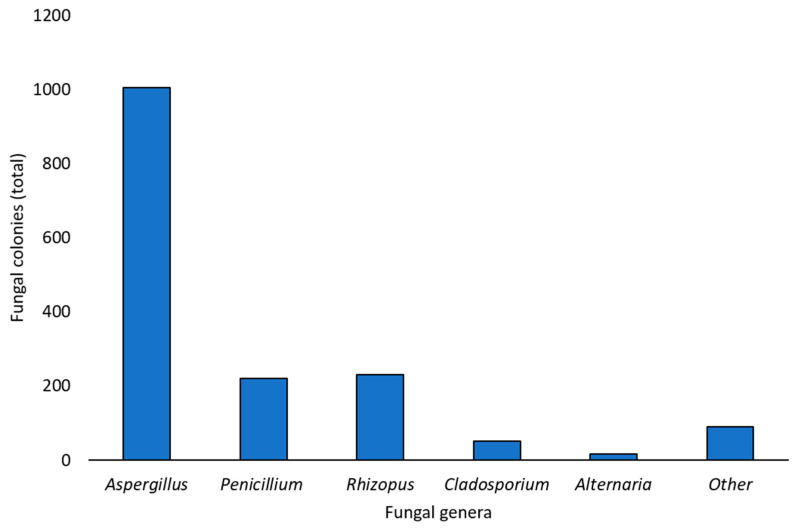
Total number of fungal colonies developed from the total marketed chickpea seed samples with all the isolation techniques used in this survey (Potato Dextrose Agar with surface disinfection, Potato Dextrose Agar without surface disinfection, deep-freezing blotter) and identified as belonging to different genera or not assigned to any genus (other) based on the observation of morphological features.

**Figure 2 foods-14-02610-f002:**
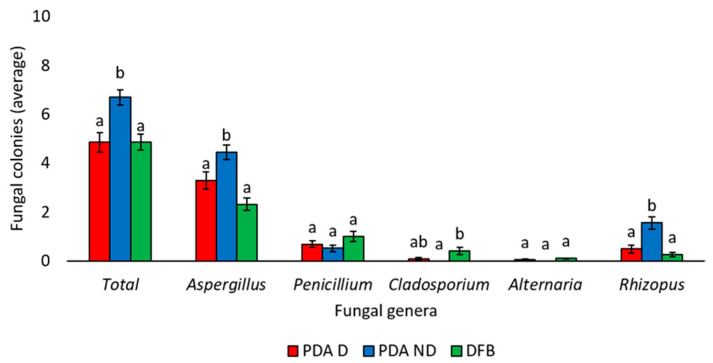
Average number of total fungal colonies or fungal colonies belonging to the *Aspergillus*, *Penicillium*, *Cladosporium*, *Alternaria,* and *Rhizopus* genera developed from marketed chickpea seed samples using three different isolation methods: Potato Dextrose Agar with surface disinfection (PDA D), Potato Dextrose Agar without surface disinfection (PDA ND), and the deep-freezing blotter (DFB). The genus of each fungal colony was morphologically identified. Each column represents the average (±standard error) of the 20 analyzed samples. Within each category (Total, *Aspergillus*, *Penicillium*, *Cladosporium*, *Alternaria*, *Rhizopus*), different letters indicate significant differences (*p* < 0.05, Tukey’s HSD test) among the three different isolation methods (PDA D, PDA ND, or DFB).

**Figure 3 foods-14-02610-f003:**
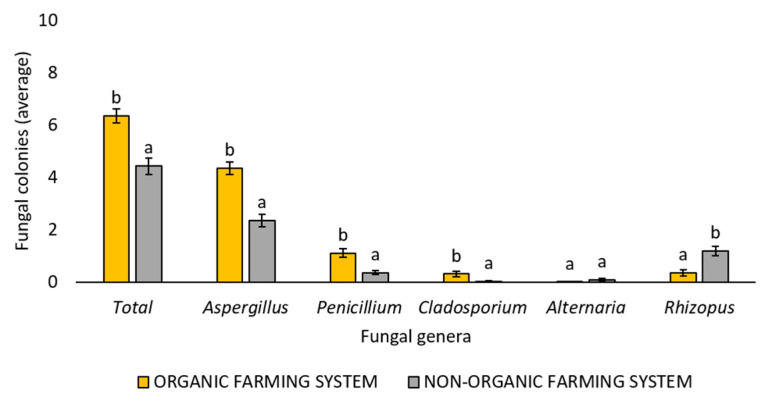
Average number of total fungal colonies or fungal colonies belonging to the *Aspergillus*, *Penicillium*, *Cladosporium*, *Alternaria,* and *Rhizopus* genera developed following the use of all three different isolation methods (Potato Dextrose Agar with surface disinfection, Potato Dextrose Agar without surface disinfection, and deep-freezing blotter) from marketed chickpea seed samples obtained with organic or non-organic farming systems. The genus of each fungal colony was morphologically identified. Each column represents the average (±standard error) of the 10 analyzed samples per each farming system. Different letters indicate significant differences (*p* < 0.05, Tukey’s HSD test) between the two different farming systems within each category (Total, *Aspergillus*, *Penicillium*, *Cladosporium*, *Alternaria*, *Rhizopus*).

**Figure 4 foods-14-02610-f004:**
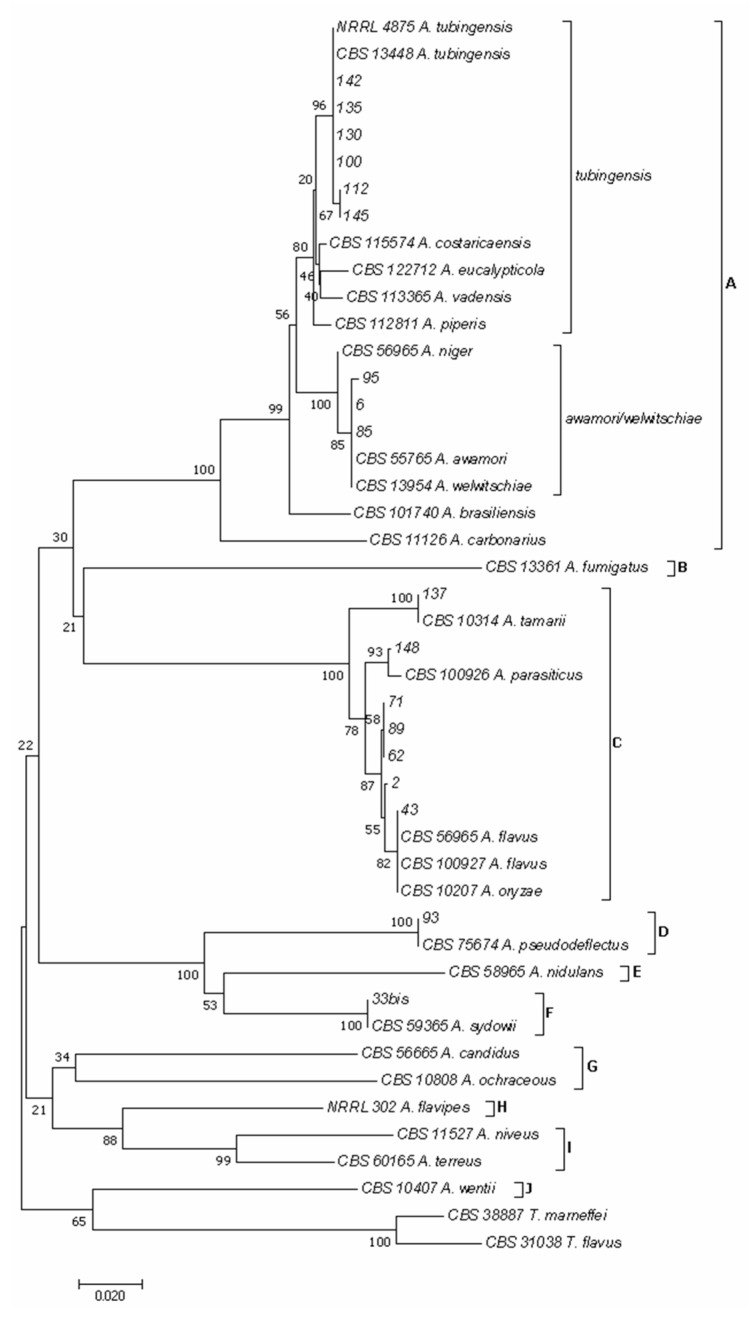
Phylogeny of *Aspergillus* isolates obtained from the marketed chickpea seed samples surveyed in this study. The evolutionary history of *Aspergillus* isolates was inferred using the Neighbor-Joining method [39] and a combined dataset of *BenA* and *CaM* gene sequences. The optimal tree with the sum of branch length = 1.35228509 is shown. The percentage of replicate trees in which the associated taxa clustered together in the bootstrap test (1000 replicates) are shown next to the branches [40]. The tree is drawn to scale, with branch lengths in the same units as those of the evolutionary distances used to infer the phylogenetic tree. The evolutionary distances were computed using the Maximum Composite Likelihood method [41] and are in the units of the number of base substitutions per site. The analysis involved 46 nucleotide sequences. All positions containing gaps and missing data were eliminated. There was a total of 451 positions in the final dataset. Evolutionary analyses were conducted in MEGA software version 7.0 [34]. In the phylogram clade A included species of the section *Nigri*, clade B of the section *Fumigati*, clade C of the section *Flavi*, clade D of the section *Usti*, clade E of the section *Nidulantes*, clade F of the section *Versicolores*, clade G of the sections *Candidi*/*Circumdati*, clade H of the section *Flavipedes*, clade I of the section *Terrei,* and clade J of the section *Cremei*.

**Figure 5 foods-14-02610-f005:**
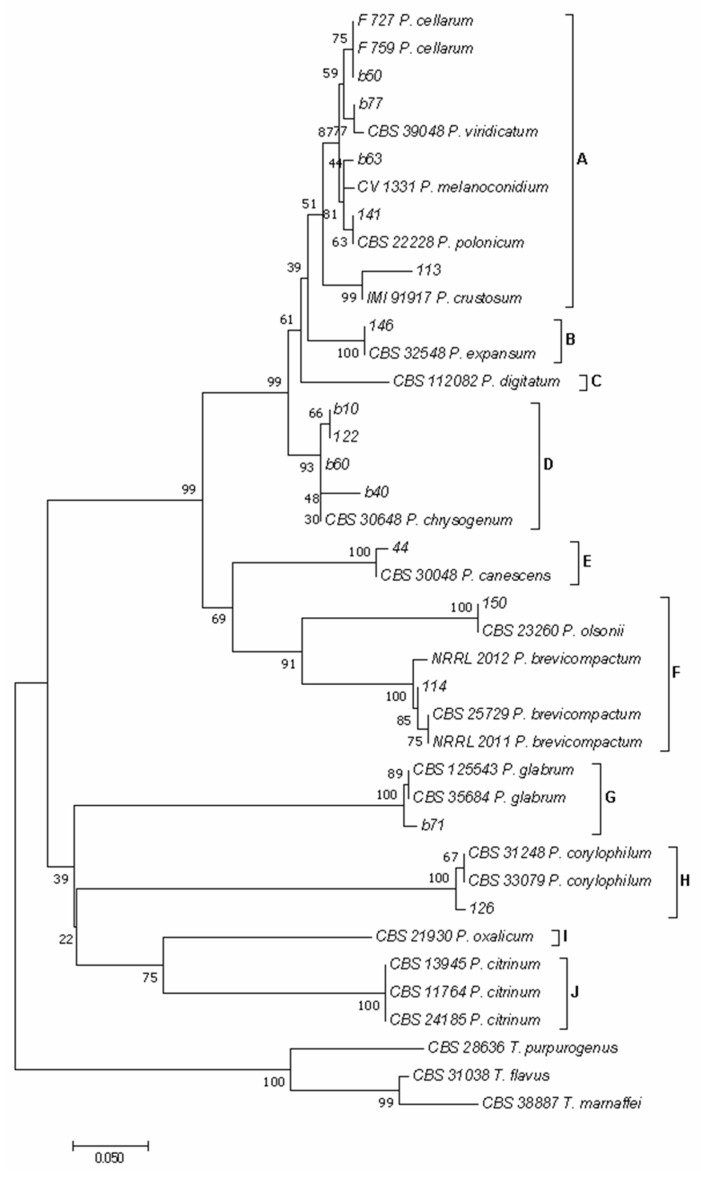
Phylogeny of *Penicillium* isolates obtained from marketed chickpea seed samples surveyed in this study. The evolutionary history of *Penicillium* isolates was inferred using the Neighbor-Joining method [39] and a combined dataset of *BenA* and *CaM* gene sequences. The optimal tree with the sum of branch length = 2.10774471 is shown. The percentage of replicate trees in which the associated taxa clustered together in the bootstrap test (1000 replicates) are shown next to the branches [40]. The tree is drawn to scale, with branch lengths in the same units as those of the evolutionary distances used to infer the phylogenetic tree. The evolutionary distances were computed using the Maximum Composite Likelihood method [41] and are in the units of the number of base substitutions per site. The analysis involved 40 nucleotide sequences. All positions containing gaps and missing data were eliminated. There was a total of 161 positions in the final dataset. Evolutionary analyses were conducted in MEGA software version 7 [34]. In the phylogram clade A included species of the section *Fasciculata*, clade B of the section *Penicillium*, clade C of the section *Digitata*, clade D of the section *Chrysogena*, clade E of the section *Canescentia*, clade F of the section *Brevicompacta*, clade G of the section *Aspergilloides*, clade H of the section *Exilicaulis*, clade I of the section *Lanata*-*Divaricata,* and clade J of the section *Citrina*.

**Figure 6 foods-14-02610-f006:**
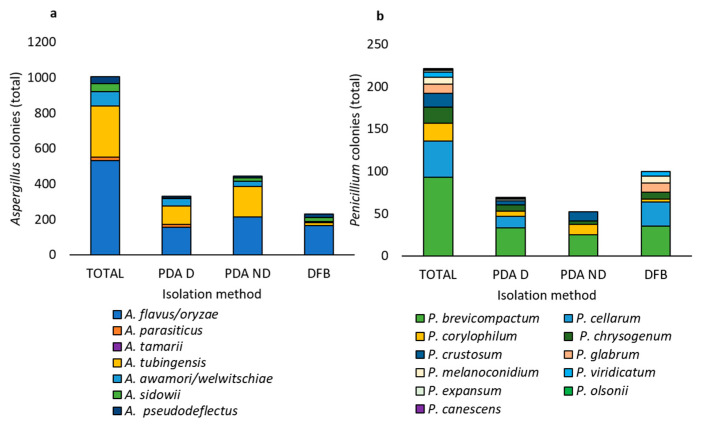
Total number of fungal colonies of *Aspergillus* (**a**) and *Penicillium* (**b**) species obtained from the marketed chickpea seed samples, as identified by phylogenetic analysis. The TOTAL category shows the sum of colonies obtained using all three isolation methods (Potato Dextrose Agar with seed surface disinfection, Potato Dextrose Agar without seed surface disinfection, and deep-freezing blotter); the PDA D category shows the colonies obtained using the Potato Dextrose Agar method with seed surface disinfection; the PDA ND category shows the colonies obtained using the Potato Dextrose Agar method without seed disinfection; the DFB category shows the colonies obtained using the deep-freezing blotter method.

**Figure 7 foods-14-02610-f007:**
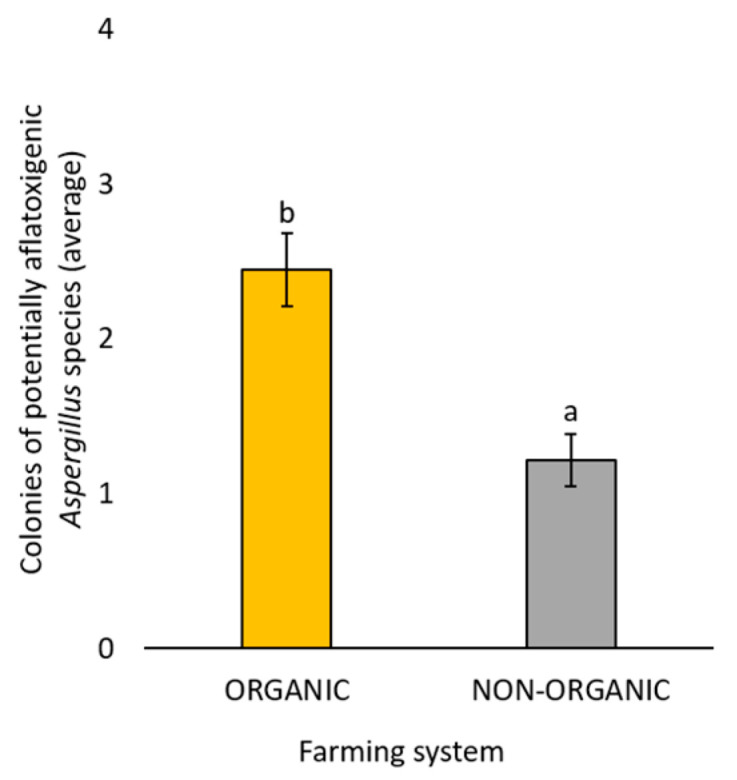
Average number of fungal colonies belonging to the potentially aflatoxigenic *Aspergillus* species (*A. flavus*/*oryzae* and *A. parasiticus*) developed following the use of all three different isolation methods (Potato Dextrose Agar with seed surface disinfection, Potato Dextrose Agar without seed surface disinfection, and deep-freezing blotter) from the marketed chickpea seed samples obtained with organic or non-organic farming systems. The species were phylogenetically identified. Each column represents the average (±standard error) of the 10 analyzed samples per each farming system. Different letters indicate significant differences (*p* < 0.05, Tukey’s HSD test) between the two different farming systems.

**Figure 8 foods-14-02610-f008:**
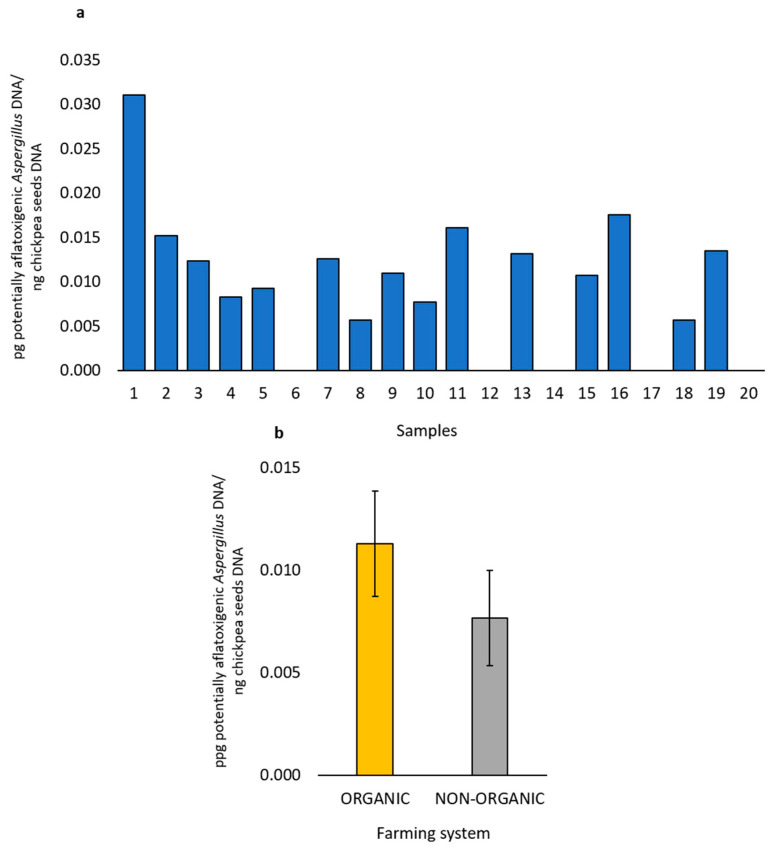
DNA amount of potentially aflatoxigenic *Aspergillus* species detected by qPCR in each of the marketed chickpea seed samples (**a**). Average amount (ten samples ± standard error) of DNA of potentially aflatoxigenic *Aspergillus* species in the marketed chickpea seed samples from organic or non-organic farming systems (**b**).

**Table 1 foods-14-02610-t001:** Marketed dried chickpea seed samples collected in 2019 and analyzed in this study.

Sample	Farming System	Origin	Packaging
1	Organic	EU (Umbria, Italy)	Sealed Plastic Bag
2	Organic	EU (Umbria, Italy)	Sealed Plastic Bag
3	Organic	EU (Umbria, Italy)	Sealed Plastic Bag
4	Organic	Extra-EU	Sealed Plastic Bag
5	Organic	EU (Umbria, Italy)	Unsealed Cardboard Box
6	Organic	EU (Umbria, Italy)	Sealed Plastic Bag
7	Organic	EU (Umbria, Italy)	Sealed Plastic Bag
8	Organic	EU (Umbria, Italy)	Sealed Plastic Bag
9	Organic	EU (unspecified)	Sealed Plastic Bag
10	Organic	EU (Umbria, Italy)	Sealed Plastic Bag
11	Non-Organic	EU (Umbria, Italy)	Sealed Plastic Bag
12	Non-Organic	EU (Puglia, Italy)	Sealed Plastic Bag
13	Non-Organic	EU (Italy)	Sealed Plastic Bag
14	Non-Organic	EU (Umbria, Italy)	Sealed Plastic Bag
15	Non-Organic	EU (Umbria, Italy)	Sealed Plastic Bag
16	Non-Organic	EU (Umbria, Italy)	Sealed Plastic Bag
17	Non-Organic	EU (Spain)	Sealed Plastic Bag
18	Non-Organic	Unspecified	Sealed Plastic Bag
19	Non-Organic	EU (The Marche, Italy)	Sealed Plastic Bag
20	Non-Organic	EU (Umbria, Italy)	Sealed Plastic Bag

**Table 2 foods-14-02610-t002:** Presence (X) of fungal colonies belonging to the genera *Aspergillus*, *Penicillium*, *Cladosporium*, *Alternaria*, *Rhizopus,* or other genera in the marketed chickpea seed samples analyzed in this study.

Fungal Genera	Samples																			
	1	2	3	4	5	6	7	8	9	10	11	12	13	14	15	16	17	18	19	20
*Aspergillus*	X	X	X	X	X	X	X	X	X	X	X	X	X		X	X	X	X	X	X
*Penicillium*	X	X		X		X			X	X	X	X	X				X	X	X	X
*Cladosporium*	X								X	X			X							
*Alternaria*	X										X		X							
*Rhizopus*	X	X	X			X	X	X			X	X	X			X	X	X	X	X
Other genera	X	X	X	X	X	X					X			X	X	X	X	X	X	

**Table 3 foods-14-02610-t003:** Fungal secondary metabolites (ng/g) detected by LC-MS/MS in the marketed chickpea seed samples analyzed in this study.

Secondary Metabolites *	Samples from Organic Farming System										Samples from Integrated Farming System									
	1	2	3	4	5	6	7	8	9	10	11	12	13	14	15	16	17	18	19	20
**Cyclobutanediones**																				
Moniliformin ^Fus^	0.24	0.14	0.56	0.14	<LOD	<LOD	<LOD	<LOD	<LOD	<LOD	<LOD	<LOD	<LOD	<LOD	<LOD	<LOD	<LOD	<LOD	<LOD	<LOD
**Cyclodepsipeptides**																				
Destruxin B ^Fus^	0.42	<LOD	0.30	<LOD	<LOD	<LOD	0.01	<LOD	<LOD	<LOD	<LOD	<LOD	0.03	<LOD	<LOD	<LOD	<LOD	<LOD	<LOD	<LOD
Enniatin A ^Fus^	<LOD	<LOD	<LOD	<LOD	<LOD	<LOD	<LOD	<LOD	<LOD	<LOD	<LOD	0.001	<LOD	<LOD	<LOD	<LOD	<LOD	<LOD	<LOD	<LOD
Enniatin A1 ^Fus^	0.02	0.01	0.04	<LOD	<LOD	<LOD	<LOD	<LOD	<LOD	<LOD	<LOD	0.04	0.01	0.01	<LOD	<LOD	0.01	<LOD	<LOD	<LOD
Enniatin B ^Fus^	0.37	0.21	0.72	0.18	0.09	0.04	0.02	0.02	0.01	0.01	<LOD	0.14	0.02	0.03	0.02	0.02	0.02	0.01	0.01	0.01
Enniatin B1 ^Fus^	0.07	0.05	0.21	0.05	0.03	0.01	0.01	0.01	<LOD	0.01	<LOD	0.09	0.01	0.02	0.01	0.01	0.01	<LOD	0.01	0.01
W493 ^Fus^	<LOD	<LOD	<LOD	<LOD	<LOD	<LOD	<LOD	<LOD	<LOD	<LOD	<LOD	0.29	<LOD	<LOD	<LOD	<LOD	0.11	<LOD	<LOD	<LOD
**Cyclopeptides**																				
Tentoxins ^Al^	<LOD	<LOD	0.30	<LOD	<LOD	<LOD	<LOD	<LOD	<LOD	<LOD	<LOD	<LOD	<LOD	<LOD	<LOD	<LOD	<LOD	<LOD	<LOD	<LOD
**Propionic acids**																				
3-Nitropropionic acid ^As^	<LOD	<LOD	<LOD	<LOD	<LOD	<LOD	<LOD	<LOD	<LOD	<LOD	<LOD	<LOD	0.88	<LOD	<LOD	<LOD	<LOD	<LOD	<LOD	<LOD
**Sesquiterpenes**																				
Sporogen AO ^As^	<LOD	<LOD	<LOD	<LOD	<LOD	0.19	<LOD	<LOD	<LOD	<LOD	<LOD	<LOD	<LOD	<LOD	<LOD	<LOD	<LOD	<LOD	<LOD	<LOD
**Tetramic acids**																				
Tenuazonic acid ^Al^	6.35	<LOD	<LOD	<LOD	<LOD	<LOD	<LOD	<LOD	<LOD	<LOD	<LOD	<LOD	6.50	<LOD	<LOD	<LOD	<LOD	<LOD	<LOD	<LOD
**Trichothecenes**																				
Deoxynivalenol ^Fus^	1.61	N.D.	2.67	0.59	0.46	<LOD	<LOD	<LOD	<LOD	<LOD	<LOD	<LOD	<LOD	<LOD	<LOD	<LOD	<LOD	<LOD	<LOD	<LOD
**Xanthone derivatives**																				
Sydowinin B ^As^	2.06	1.54	3.54	1.93	1.62	1.04	3.94	0.76	1.15	0.97	1.32	1.63	2.14	1.47	1.90	1.51	3.44	2.37	1.18	1.32

* The attribution of a compound to a specific chemical category was assessed according to [50,51,52,53,54]; ^Al^
*Alteranaria* secondary metabolites; ^As^
*Aspergillus* secondary metabolites; ^Fus^
*Fusarium* secondary metabolites; LOD, limit of detection.

**Table 4 foods-14-02610-t004:** Secondary metabolites (ng/g) detected in the in vitro cultures of potentially aflatoxigenic isolates belonging to the *Aspergillus* species isolated from the marketed chickpea seed samples and grown on Czapek Yeast Autolysate agar medium.

Secondary Metabolites *	*Aspergillus flavus*/*oryzae*					*Aspergillus parasiticus*
	2	43	62	71	89	148
**Aflatoxins**						
Aflatoxicol	<LOD	<LOD	<LOD	<LOD	<LOD	33.9
Aflatoxin B1	<LOD	<LOD	<LOD	<LOD	<LOD	18,100
Aflatoxin B2	<LOD	<LOD	<LOD	<LOD	<LOD	6640
Aflatoxin G1	<LOD	<LOD	<LOD	<LOD	<LOD	30,800
Aflatoxin G2	<LOD	<LOD	<LOD	<LOD	<LOD	6050
Aflatoxin M1	<LOD	<LOD	<LOD	<LOD	<LOD	752
*O-*Methylsterigmatocystin	<LOD	<LOD	<LOD	<LOD	<LOD	214
Sterigmatocystin	<LOD	<LOD	<LOD	<LOD	<LOD	72.5
**Anthraquinoids**						
Averufin	<LOD	<LOD	<LOD	<LOD	<LOD	962
Norsolorinic acid	<LOD	<LOD	<LOD	<LOD	<LOD	210
**Depsipeptides**						
Aspergillicin derivatives	441	<LOD	<LOD	<LOD	<LOD	3730
**Dihydrobenzofuran derivatives**						
Asperfuran	159	215	47.4	65.1	58	<LOD
**Isocumarins**						
Dichlordiaportin	3.72	<LOD	<LOD	0.94	0.59	187
*O-*Methyldichlordiaportin	<LOD	<LOD	<LOD	<LOD	<LOD	43.7
**Koningic acids**						
Heptelidic acid	<LOD	683	1220	673	521	<LOD
**Monoterpenoids**						
Gliocladic acid	<LOD	4440	2750	3990	4190	<LOD
**Penicillins**						
Penicillin G	123,000	1,390,000	1,850,000	961,000	373,000	<LOD
**Propionic acids**						
3-Nitropropionic acid	2570	4960	26,200	10,100	12,200	463,000
**Pyrones**						
Kojic acid	33.5	676	1010	687	77.5	401,000
**Sesquiterpenes**						
Sporogea AO	152.5		67	55.4	19.9	<LOD
**Steroids**						
Helvolic acid	<LOD	<LOD	<LOD	<LOD	<LOD	169
**Terpenes**						
Cyclopiazonic acid	<LOD	62,600	<LOD	47,200	38,500	<LOD
**Versicolorins**						
Versiconal acetate	<LOD	<LOD	<LOD	<LOD	<LOD	34.8
Versicolorin C	<LOD	<LOD	<LOD	<LOD	<LOD	1060
**Others**						
NP 1243	2.72	1.73	<LOD	1.39	1.37	<LOD

* The attribution of a compound to a specific chemical category was assessed according to [53,55,56]; LOD, limit of detection.

**Table 5 foods-14-02610-t005:** Number of colonies of potentially aflatoxigenic *Aspergillus* species isolated and DNA amount of potentially aflatoxigenic *Aspergillus* species as detected and quantified by qPCR from each marketed chickpea seed sample analyzed in this study.

	Samples																			
	1	2	3	4	5	6	7	8	9	10	11	12	13	14	15	16	17	18	19	20
**Potentially aflatoxigenic *Aspergillus* species detected by isolation methods (number of colonies)**																				
*A. flavus*/*oryzae*	49	61	86	39	71	1	69	-	-	-	10	6	12	-	39	42	-	27	-	10
*A. parasiticus*	-	-	-	-	-	-	-	-	-	-	-	-	-	-	-	-	-	-	17	-
TOTAL	49	61	86	39	71	1	69	-	-	-	10	6	12	-	39	42	-	27	17	10
**Potentially aflatoxigenic *Aspergillus* species detected by qPCR (pg *Aspergillus* DNA/ng chickpea seeds DNA)**																				
TOTAL	0.031	0.015	0.012	0.008	0.009	<LOD	0.013	0.006	0.011	0.008	0.016	<LOD	0.013	<LOD	0.011	0.018	<LOD	0.006	0.014	<LOD

LOD, limit of detection.

## Data Availability

The original contributions presented in the study are included in the article/Appendix A, further inquiries can be directed to the corresponding author.

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
