# Peer review of "Occurrence of Aspergillus and Penicillium Species, Accumulation of Fungal Secondary Metabolites, and qPCR Detection of Potential Aflatoxigenic Aspergillus Species in Chickpea (Cicer arietinum L.) Seeds from Different Farming Systems"

_foods, 2025, doi:10.3390/foods14152610_

Round 1
Reviewer 1 Report
Comments and Suggestions for Authors
The manuscript focuses on the monitoring of mycotoxigenic moulds in chickpeas. While the research topic is relevant and potentially valuable, the manuscript lacks clarity in several key areas, making it difficult to follow.
Title and Abstract
The title is unclear, particularly due to the inappropriate use of the term “pipeline,” which seems out of context in this case. A more precise and scientifically appropriate title is recommended.
The abstract is overly long and does not effectively introduce the main problem or rationale behind the study. It would benefit from a clearer statement of the research question and the significance of monitoring mycotoxigenic fungi in chickpeas.
The reason for choosing "organic" and "integrated" chickpeas should be better explained. Consider whether terms such as "conventional" or "industrial" might be more appropriate or widely understood in the context of agricultural production systems.
Materials and Methods
The description of the chickpea samples is confusing. It is unclear how many types were used, how they differ, and how they were selected or sourced. A clearer presentation of sample types and study design is necessary.
The mycological methods using PDA-ND and PDA-D media are well described, but the DFB medium is only referenced (1966), and the source is not openly accessible. At least a brief description of this method should be included to ensure reproducibility.
Similarly, the sample preparation steps for LC-MS/MS analysis are not sufficiently described and should be detailed to allow others to replicate the study.
Results and Discussion
The results section is thorough and well presented.
However, the discussion is underdeveloped. The authors should provide a deeper interpretation of their findings, relate them to existing literature, and elaborate on the implications for food safety, agriculture, or future research.
Comments on the Quality of English LanguageSome of the terms are very unclear; therefore, it is hard to follow the text. English should be improved.
Author Response
Dear Reviewer 1,
please find the attached file.

Reviewer 2 Report
Comments and Suggestions for Authors
- The introduction should supplement fungal contamination monitoring data from major chickpea consumption regions (e.g., Europe) to strengthen the research necessity. Clearly propose a mechanistic hypothesis linking the ban on synthetic pesticides in organic systems to fungal risks (e.g., limitations of biocontrol).
- Materials and Methods:
- What is the representativeness of 20 commercial dried samples? Since the sample sources only cover Perugia, Italy, lacking comparison with other major European producing countries, the regional limitations should be clarified.
- How were confounding factors such as climate, season, and storage conditions (temperature/humidity) affecting mycotoxin production excluded?
- What is the origin of the "Unspecified" sample? Does it meet the inclusion criteria? It is recommended to supplement Table 1 with the sample collection year and storage conditions (temperature/humidity).
- Discussion:
- Emphasize the capture efficiency of fungal diversity by the combined use of three methods, rather than merely describing differences, and propose a strategic plan for integrating the three methods.
- Although Aspergillus colony counts were significantly higher in organic samples, the study did not analyze their correlation with agronomic practices (e.g., crop rotation, irrigation). It is suggested to deeply discuss the impact mechanisms of agricultural management measures on fungal communities.
Author Response
Dear Reviewer 2,
please find the attached file.

Round 2
Reviewer 1 Report
Comments and Suggestions for Authors
It can be seen that authors improved the manuscript according to comments.
Just one comment: in the sentence: Contamination by mycotoxigenic fungi and mycotoxins is poorly investigated in chickpeas grown in Europe, in which fungi of the mycotoxigenic Aspergillus and Penicillium genera and their related mycotoxins, such as aflatoxins B1, B2, G1 and G2, have been detected [17], (lines 69-72), please add OTA, beacuse Penicillium species don't produce AF.
Author Response
Thanks for your comment, we addressed it by adding OTA to the sentence.